# OpenReview forum: "Under the Influence: Quantifying Persuasion and Vigilance in Large Language Models"
_ICLR.cc/2026/Conference — Submitted to ICLR 2026_

### Official Review · Reviewer_QCSx · 2025-10-28

**Soundness:** 2
**Presentation:** 3
**Contribution:** 2
**Rating:** 4
**Confidence:** 4

**Summary:**

Given the lack of systems to examine the social capabilities of large language models (LLMs), the paper considers investigating the task performance, persuasion and vigilance of frontier LLMs. They first build a game environment upon Sokoban, where LLMs can take the roles of players or advisors. In addition, the advisor can either provide benevolent or malicious advice, while the player may or may not be given the hint of the existence of misleading advice. Then quantifiable definitions for performance, persuasion and vigilance are proposed. Finally, five frontier models are examined w.r.t. such metrics with follow-up empirical analysis of resource rationality and persuasion strategies.

**Strengths:**

> **Originality**
- The paper provides a new game environment to examine performance, persuasion and vigilance.

> **Quality**
- Several frontier models are tested in the performance comparisons.

**Weaknesses:**

> **Quality**
- The definition of persuasion: Given $z_i(M_A)=\omega=1$, the score is always zero. However, the included two cases are different. For the case when $z_i(M_A|M_B^\omega)=\omega$, the interpretation of the scenario is that

    ''$M_B^\omega$ cannot further improve the decision of $M_A$ as $M_A$ is already capable of solving the problem i''.

    However, if $z_i(M_A)=\omega=1$ and $z_i(M_A|M_B^\omega)\neq\omega$, the interpretation is that

    ''B's persuasion even makes A less confident to keep the correct choice of solution.''

    Letting the score take same value, zero, in these two cases may not yet show the capability of B in a finer sense.

- A similar concern can be found in the definition of vigilance: e.g., given $z_i(M_A|M_B^\omega)=1$ and $\omega\neq1$, consider

    ''$z_i(M_A)=1$'' v.s. ''$z_i(M_A)\neq1$''.

    Feel free to correct the reviewer in case of misunderstanding the definitions.

> **Clarity**
- Section 3.3.2: The average solve rate case could be explicitly provided with notations.
- Equation (2)-(3): A quick explanation of the denominators (renormalization) can be helpful.
- Equations (2)-(3): This can be a minor but necessary to mention point that usually the denominators do not vanish.
- Line 272-275: The rationale behind different scenarios could be explained further.

> **Significance**
- The properties of the evaluation paradigm being ''scalable'' or ''rich'' are not explicitly justified.

**Questions:**

- What is the motivation, in Equation (3), that cases are summed in numerators and denominators respectively, instead of taking the average of the two ratios ($\omega=0, 1$)?
- The same question as the above for equation (5).
- Could further details on the calculation of ''optimal ratio'' in Figure 3 be provided?
- For the discussion in Line 323, why a planner would guarantee such independence?

---

> ### Author Response · Authors · 2025-11-27
> **Author Response Part 1**
>
> Thank you for your valuable feedback\! We address your comments below.
>
> **W1 Metrics:**
> For persuasion, you are absolutely correct that the first case indeed refers to trials where we are unable to measure persuasiveness due to the player already solving the problem, and the second case refers to trials where the persuasive model is so bad that it actually persuades the player to the opposite of the persuader’s objective. However, our persuasion rate metrics (Eqns 2 and 3\) account for this by normalizing the number of trials where the persuader is successful by the number of trials where there is signal for measuring persuasiveness. In other words, in the denominator, we exclude all trials of the first case where the player model was already doing the behavior desired by the persuasive model. As a result, persuaders are not penalized for the first case, only for the second case. We similarly normalize for the vigilance rate metrics (see denominators in Eqns 4 and 5). We apologize for the confusion, we have updated the metrics section to clarify this.
>
> **W2 Clarity:**
> Thank you for the suggestions. As mentioned above, we have clarified this section, and implemented these suggestions as part of that. In particular, we hope that the summary above clarifies the point about renormalization, and it is indeed the case that these denominators rarely vanish, though we note that this can happen for some of the metrics in certain extreme cases (e.g. vigilance with beneficial advice could not be measured for GPT5 because it saturated performance across all runs on all ten puzzles). For the case of vigilance (lines 272-275), at a high level, we measure vigilance as “the number of trials where a player ignores bad advice or follows good advice, minus the number of trials where a player follows bad advice or ignores good advice, divided by the number of trials where signal is actually measurable (i.e. the denominator excludes trials where the unassisted player already was doing the action desired by the assistant since we cannot tell if persuasion has any effect in these cases)”. We have updated this section accordingly to make it easier to interpret.
>
> **W3 Significance:**
> We chose Sokoban intentionally as it is a tightly-controlled, tractable environment which allows us to isolate research findings from confounding factors that may be found in more “realistic” settings. Sokoban specifically has several nice features for studying persuasion and vigilance: it involves sequential decision-making (unlike the vast majority of persuasion studies which focus on single decisions), it has multiple mechanisms for failure (deadlocking states, sub-optimal planning), and it can be ratcheted up or down in complexity easily by looking at planning complexity from symbolic planners.
>
> This allowed us to conduct multiple more fine-grained analyses of persuasion, such as investigating what kinds of deceptive strategies models employ to trick a player (deadlocking vs. suboptimality). We expect that follow-up work in this area will extend our findings into even richer domains. Are there any particular settings which the reviewer would like to see in an additional experiment? We are happy to try to run these experiments before the final rebuttal deadline.

---

> ### Author Response · Authors · 2025-11-27
> **Author Response Part 2**
>
> **Q1 and Q2 Equation motivations:**
> This is to account for imbalance in the number of occurrences of each case (e.g. a model that has very high unassisted performance may only have very few trials where vigilance for beneficial persuasion is measurable but many where vigilance against malicious persuasion is measurable).
>
> **Q3: Optimal ratio**
> The optimal ratio was computed as the number of times each player model’s next move was the same as the next optimal move computed by the shortest planner solution, divided by the number of total moves. For example, if the next action in the shortest planner solution was DOWN, we would increment the numerator if the player also chose DOWN, and increment the denominator regardless. We can do this as the optimal planner solution is recomputed after each move. For example, in Figure 3, GPT-5 chooses this optimal move \~90% of the time.
>
> **Q4: Access to a planner for capability measurement independence**
> Access to a planner ensures that what we are measuring is the persuader’s ability to persuade, rather than their ability to solve the underlying task itself. For example, DeepSeek is mediocre at the game in the unassisted case (\~6/10) but still manages to improve the performance of most other models in the beneficial assistant setting, and lower the performance of most other models in the malicious assistant setting, when given access to the planner.
>
> That being said, we have now conducted an additional supplementary experiment to determine persuasive ability without access to the planner. Within our relatively small experiment design (379 moves, 5 players models, 5 advisor models, 1 puzzle), we compared the optimal move ratio (described above) of each player between the unassisted and malicious case. We chose this metric over puzzle solve rate due to a lack of structure in how to divide responses without access to the planner and the reduced number of experiments. We find similar trends to the experiments with access to the planner: all player model’s performance degrades substantially (although less than with access to the planner), GPT-5 continues to be the most persuasive and vigilant model, and Grok 4 Fast continues to be unpersuasive and tricked into deadlocks and suboptimal plans despite its relatively good unassisted performance. See our new Appendix Section A.3 for more details.

---

### Official Review · Reviewer_nSG7 · 2025-10-28

**Soundness:** 2
**Presentation:** 3
**Contribution:** 3
**Rating:** 4
**Confidence:** 3

**Summary:**

The paper introduces a controlled evaluation framework to measure persuasion, vigilance, and task performance in large language models through a Sokoban-based setup involving benevolent and malicious advisors. Five frontier models are tested under various advice conditions, revealing that strong performance does not necessarily imply either persuasion or vigilance, and that models differ widely in susceptibility to persuasive manipulation. Overall, the paper targets an important safety and reliability dimension of LLM behaviour.

**Strengths:**

The paper introduces and thoroughly evaluates an important safety and reliability concept, the separation between task competence, persuasion, and vigilance. The experimental design, metrics, and multi-model evaluation together make this a meaningful contribution toward understanding LLM robustness to external influence.

**Weaknesses:**

- The presentation of results and metric definitions is somewhat difficult to follow. I found myself going back and forth between sections to connect the definitions with the numbers in the tables, and I am still not entirely confident I understand the results shown (particularly in Table 1).
- The generalisation of these findings beyond the specific Sokoban setup is not clear. Additionally, It would be informative to see results without access to the “gold” planner solutions.

**Questions:**

1. In Table 1, Claude Sonnet 4 achieves a vigilance score of 0.087 for benevolent advice. Since the metric ranges from −1 to 1, this suggests the model only barely benefited from good advice. Could the authors clarify how to interpret such small positive values in practical terms, especially since they state “Every player achieves close to ceiling performance when paired with a benevolent LLM advisor”.
2. There appears to be a missing data point in Table 1. Can the authors confirm whether this was intentional or due to evaluation issues?
3. While some statistical tests are reported, others are missing. Would the authors consider adding statistical significance or confidence intervals for all main results to strengthen the conclusions?
4. It would be useful to see whether results hold without advisors having access to the planner’s gold solution, or under weaker/noisy advisors.
5. How well do the persuasion and vigilance rankings generalise to other sequential tasks or domains beyond Sokoban?

---

> ### Author Response · Authors · 2025-11-27
>
> Thank you for your valuable feedback\! We address your comments below.
>
> **W1 Metric definitions rewriting**
> Thank you for bringing this up, we have updated the metrics section to make it clearer and easier to follow. To summarize at a high level: we measure persuasion as “the proportion of trials where an advisor persuades a player to change their behavior in the desired direction (i.e., if the advisor is malicious then this counts the proportion of trials where the player solved the puzzle when unassisted but now fails to solve it, if the advisor is benevolent then it counts the proportion of trials where the player previously failed the puzzle but now solves it) out of the number of trials where signal is actually measurable (i.e. the denominator excludes trials where the unassisted player already was doing the action desired by the advisor since we cannot tell if persuasion has any effect in these cases), and we measure vigilance as “the number of trials where a player ignores bad advice or follows good advice, minus the number of trials where a player follows bad advice or ignores good advice, divided by the number of trials where signal is actually measurable (i.e. the denominator excludes trials where the unassisted player already was doing the action desired by the assistant since we cannot tell if persuasion has any effect in these cases)”. We have updated the section accordingly.
>
> **W2 / Q4 / Q5 Generalization of findings**
> Thank you for asking about the generalization of the results. We now include experiments in the malicious persuasion case where advisors *do not have access to a gold planner* in Appendix section A.3. We find broadly similar results – namely that models can understand the puzzle very well and still not be persuasive advisors. For example, Grok-4-fast performs very well in the unassisted case, but does not have a strong negative impact on player model performance.
>
> For generalization experiments, could the reviewer suggest a particular setting they would find compelling? We can attempt to generate results before the final rebuttal deadline.
>
> **Q1** **Interpretation of vigilance scores**
> As explained above, our vigilance score increases if a model listens to benevolent advice but decreases if it ignores it (vice versa for malicious advice). In the case of Claude as a player, as you can see in Figure 4, it solves 3 puzzles when unassisted, and then solves some, but not all, additional puzzles when given benevolent advice. The low score it receives is thus a result of it following beneficial advice roughly as often as it ignored it. We have updated the sentence “Every player achieves close to ceiling performance when paired with a benevolent LLM advisor” to match the correct statement we had previously written in the caption of Figure 4, which was “When advice is benevolent, most models perform near ceiling…”. Claude is a notable exception to this (though it does perform quite well when advised by Gemini 2.5 Pro).
>
> **Q2 Missing data point for GPT-5**
> The missing data point is due to GPT-5 performing perfectly across all 5 repetitions on each of the 10 puzzles we considered. As a result, it is already at ceiling performance and there were no trials where we could test whether it benefited from benevolent advice.
>
> **Q3 Adding statistical tests by end of the week**
> We will add the remaining statistical significance tests for all reported results by the end of this week. **Update**: we have added the significance tests in a comment below.

---

> > ### Author Response · Authors · 2025-12-03
> > **Remaining statistical tests**
> >
> > As mentioned before, we are including all relevant claims and their statistical significance below.
> >
> > (1) "This suggests that performance, and persuasion & vigilance, are not necessarily correlated capabilities for frontier LLMs"
> >
> > A linear mixed effects model of performance (fixed effects: persuasion score, vigilance score; random effects: puzzle, model) produced:
> > persuasion: (t(44) = -0.26, p = .796, beta = -0.04, 95% CI [-0.33, 0.25])
> > vigilance: (t(45) = -0.99, p = .328, beta = -0.08, 95% CI [-0.22, 0.07])
> >
> > (2) “LLMs spend less computation (on average) when the provided advice is beneficial relative to their playing unassisted"
> >
> > A paired differences t-test between benevolent case token consumption and unassisted case token consumption produced: (t(49) = 3.241, p = .002, 95% CI [358.19, 1524.31]).
> >
> > (3) "Similarly, all LLMs spend more computation in the malicious and aware malicious conditions, suggesting that they are sensitive to malicious persuasion."
> >
> > After further statistical analysis, we found that this claim was slightly more nuanced and are rewriting it as follows:
> > “If they successfully solve a puzzle unassisted, in order to still solve it under malicious persuasion, models need to expend more compute (malicious: t(91) = 6.92, p < .001, M = 0.161, 95% CI [0.12, 0.21]; aware malicious: t(128) = 12.5, p < .001, M = 0.177, 95% CI [0.15, 0.21]). When models already fail at a puzzle when unassisted, they listen to the malicious advisor and expend fewer tokens for that puzzle (malicious: t(28) = -4.87, p < .001, M = -0.646, 95% CI [-0.92, -0.37]]; aware malicious: t(28) = -3.58, p = .001, M = -0.436, 95% CI [-0.685, -0.187]). In some cases, models that can solve puzzles on their own, fail to solve them under malicious advice, and in these cases they generally expend fewer tokens as well (malicious: t(127) = -7.01, p < .001, M = -0.498, 95% CI [-0.64, -0.36]; aware malicious: t(90) = -6.02, p < .001, M = -0.450, 95% CI [-0.60, -0.30]). Taken together, these results suggest that vigilance in the face of malicious advice requires additional compute.” Conceptually, this can be interpreted as meaning that being vigilant requires expending additional tokens (which we see in the cases where the models manage to win despite malicious advice), but often, models do not even realize they are being tricked so they listen to the persuader as if it were beneficial and accordingly use far fewer tokens.
> >
> > (4) "GPT-5 consistently uses the deadlocking hint strategy, which may be the most effective."
> >
> > A linear mixed effects model of win rate (fixed effects: deadlock proportion; random effects: puzzle, model) produced (t(48) = -3.75, p < .001, beta = -0.294, 95% CI = [-0.451, -0.136]).

---

### Official Review · Reviewer_NJnC · 2025-10-30

**Soundness:** 3
**Presentation:** 2
**Contribution:** 2
**Rating:** 4
**Confidence:** 3

**Summary:**

This paper tries to measure persuasion and vigilance in LLMs. It uses the Sokoban game, where one LLM (advisor) gives advice to another LLM (player). The advisor can be helpful (benevolent) or malicious. The paper finds that a model's ability to solve the puzzle, its ability to persuade, and its ability to ignore bad advice are all separate skills. It also shows that models use more tokens when they get malicious advice.

**Strengths:**

This paper focuses on this specific phenomenon, proposes an evaluation, and shows some interesting results. The finding that performance and vigilance are not connected is a good insight. I also thought the finding about "resource-rationality" was interesting, where models "think" harder (use more tokens) when they get malicious advice, even if they still fail. The Sokoban setup is a good, controllable way to test this.

**Weaknesses:**

My main concern is that this Sokoban setup, while "tractable", is a somewhat constrained problem. I find it hard to believe that results from pushing boxes in a grid will tell us much about persuasion in "high-stakes" areas like medicine or finance, which the paper claims to motivate its work. The "persuasion" here seems to be a few lines of text about game moves. In this case,  how would it reflect the complex, emotional, or high-stakes human decision-making in the real world?

The paper also highlights performance, persuasion, and vigilance being "dissociable". The "advisor" models were given the optimal solutions from a planner. This means the advisor's "performance" on the task (solving Sokoban) is irrelevant; it's only testing their ability to translate a plan into words. Of course, its persuasion skill is separate from its (untested) solving skill. The "player" model's unassisted performance is tested, but its "vigilance" seems to just be a measure of sycophancy, especially when the "aware" prompt is not used.

I would also encourage the authors to do a better literature review on existing works of LLM agents under influence (e.g., https://openreview.net/forum?id=KI1WQ6rLiy).

**Questions:**

See weakness

---

> ### Author Response · Authors · 2025-11-27
>
> Thank you for your valuable feedback. We are glad you found our work “interesting” and producing a “good insight,” in a setting that “is a good, controlled way to test” the phenomenon we explored. We especially appreciate your recognition of the interestingness of the resource-rational findings; we did too and think it will be of value to the broader community.
>
> **W1: Generalization to more complex, “high-stakes” areas**
> We agree with the reviewer that it would be a mistake to generalize directly from our findings to real-world cases, where social cognition is much more complex. However, our contributions lay the foundations for how persuasion and vigilance can be studied in these domains, by proposing metrics and a flexible paradigm that can be used to explore performance in those domains. Our results can be seen as providing a controlled window on the social cognition capabilities of these models. In general, we expect that moving away from well-controlled experiments to real-world settings is likely to change both persuasion and vigilance performance, but unlikely to affect the core finding from this paper: that performance, persuasion, and vigilance can be disentangled.
>
> **W2: Vigilance measures and sycophancy**
> To be vigilant requires more than not being sycophantic. While sycophancy entails alignment with the perceived intent or desires of the user, a model that is not sycophantic can simply be insensitive to these intentions, while being vigilant entails drawing calibrated inferences from a user’s requests given their intentions---a model that simply ignores a user is not vigilant; and this is reflected in our measures: player agents score highly on vigilance if they selectively ignore malicious advice \*and follow beneficial advice\* (conversely models receive low vigilance scores if they listen to malicious advice \*or ignore beneficial advice\*).
>
> **W3: Reliance on symbolic planners**
> We note that our use of the planner was aimed at decoupling persuasion and unassisted performance, but a model’s vigilance is independent of the planner, and it is therefore still interesting that vigilance and unassisted performance are disentangled. However, as part of the rebuttal, we have also run a revised version of the malicious experiment where access to a symbolic planner is not provided to the advisor model. We find broadly similar results (please see the updated Appendix section A.3), underscoring the main findings from the paper irrespective of access to a symbolic planner.
>
> Specifically, within our relatively small experiment design (379 moves, 5 players models, 5 advisor models, 1 puzzle), we find that all player models’ performance degrades substantially (although less than with access to the planner), GPT-5 continues to be the most persuasive and vigilant model, and Grok-4 continues to be unpersuasive and tricked into deadlocks and suboptimal plans despite its relatively good unassisted performance.
>
> **W4: Literature review**
> Thank you for bringing this paper to our attention. We have now incorporated it into our related work section. However, we would like to note that our paper occupies a different space from the referenced work (see Table 1 of \[1\]). Specifically we are looking at LLM-LLM interactions, not human-LLM interactions. Additionally, a major piece of our work here is about vigilance, which is not captured by the referenced work.
>
> \[1\] Zhou et al. HAICOSYSTEM: An Ecosystem for Sandboxing Safety Risks in Human-AI Interactions. COLM (2025).

---

### Official Review · Reviewer_5WYM · 2025-11-06

**Soundness:** 3
**Presentation:** 2
**Contribution:** 2
**Rating:** 4
**Confidence:** 3

**Summary:**

This paper introduces a controlled framework using the Sokoban puzzle game to study two social capacities of LLMs—persuasion (the ability to influence) and vigilance (the ability to resist misleading input). The authors evaluate five frontier models by letting them act both as “advisors” and “players.” Results reveal that persuasion, vigilance, and problem-solving skills are distinct abilities; strong reasoning does not ensure resistance to deception. The work provides the first formal quantification of these capacities and suggests monitoring them separately for AI safety.

**Strengths:**

The dual focus on persuasion and vigilance within a single evaluation setting is conceptually original and relevant for AI safety research. The Sokoban-based setup is tractable and reproducible, allowing precise measurement of LLM influence under benevolent or malicious advice. The study covers multiple leading LLMs across different conditions, using quantitative and qualitative analyses. Results highlight that persuasion and vigilance can diverge, revealing non-trivial social cognition behaviors among models.

**Weaknesses:**

1. Sokoban is more like a toy environment. it remains unclear whether findings generalize to real-world social or linguistic persuasion tasks. The evaluation uses a small puzzle set and relies on a symbolic planner for advisors.

2. All experiments are LLM-vs-LLM interactions, so it’s uncertain how these results translate to human-AI scenarios.

3. The paper identifies vulnerabilities but provides little guidance on improving vigilance mechanisms.

**Questions:**

1. Could future studies incorporate human participants to validate LLM-vs-human persuasion differences?

2. To what extent does reliance on a planner influence persuasion effectiveness, would purely autonomous LLM advisors behave differently?

3. Did the models’ reasoning traces or language use reveal why some were more vigilant than others?

4. can you provide more insights on improving vigilance mechanisms.

---

> ### Author Response · Authors · 2025-11-27
>
> Thank you for your valuable feedback. We are glad that you found our framing of studying persuasion and vigilance simultaneously to be “conceptually original” and “relevant for AI safety research”. We also appreciate that the reviewer recognized that our setup was “tractable and reproducible” and appreciated the breadth of models we tested. We address your comments below.
>
> **W1: Sokoban is a toy environment**
> We chose Sokoban intentionally as it is a tightly-controlled, tractable environment which allows us to isolate research findings from confounding factors that may be found in more “realistic” settings. Sokoban specifically has several nice features for studying persuasion and vigilance: it involves sequential decision-making (unlike the vast majority of persuasion studies which focus on single decisions), it has multiple mechanisms for failure (deadlocking states, sub-optimal planning), and it can be ratcheted up or down in complexity easily by looking at planning complexity from symbolic planners.
>
> This allowed us to conduct multiple more fine-grained analyses of persuasion, such as investigating what kinds of deceptive strategies models employ to trick a player (deadlocking vs. suboptimality). We expect that follow-up work in this area will extend our findings into more realistic domains.
>
> **W2: LLM-vs-LLM interactions only**
> In this study, we were specifically interested in LLM-vs-LLM interactions. How vigilant LLMs are is of critical importance to understanding whether and how they might be misled. For example, in online ads, AI agents need to be vigilant to not fall for scams, while also being persuasive to generate effective ads \[1\]. While it would be separately interesting to understand to what extent LLMs are persuasive to humans, this has been examined in detail in several recent papers \[2, 3, 4\], and is not the focus of our work here.
>
> \[1\] STÖCKL, A., AND NITU, J. Are AI agents interacting with online ads? arXiv preprint 2504.07112 (2025).
> \[2\] Bai et al. LLM-generated messages can persuade humans on policy issues. Nature Communications (2025).
> \[3\] Costello et al. Durably reducing conspiracy beliefs through dialogues with AI. Science (2024).
> \[4\] Zhou et al. HAICOSYSTEM: An Ecosystem for Sandboxing Safety Risks in Human-AI Interactions. COLM (2025).
>
> **W3 / Q4: Little guidance on improving vigilance mechanisms**
> Here we were interested in whether models could be vigilant by recognizing when an advisor is giving bad advice. However, our “malicious-aware” case is a simple way of improving vigilance in models. We found that when players are explicitly told that the advice may possibly be malicious, a subset of them start to ignore that advice (particularly Gemini 2.5 Pro and Grok 4 Fast). Investigating a wider suite of vigilance improving mechanisms would be an interesting avenue for future work.
>
> **Q1: Could future studies incorporate human participants to validate LLM-vs-human persuasion differences?**
> Absolutely\! However, our study is specifically about LLM-vs-LLM interactions, and we believe that LLM-vs-human interactions would add significantly more content to the paper which would be outside its scope here. This is an excellent direction for future work, as you highlight in your question.
>
> **Q2: To what extent does reliance on a planner influence persuasion effectiveness, would purely autonomous LLM advisors behave differently?**
> Based on your suggestion, in a new supplemental experiment, we tested the ability of models to be persuasive ***without*** access to a planner. We find broadly similar results, with persuasive abilities maintaining their same rank order, but being overall slightly less effective without access to the planner. Please see the new Appendix section A.3 for details.
>
> **Q3: Did the models’ reasoning traces or language use reveal why some were more vigilant than others?**
> Based on our initial analyses, we could not find patterns in the reasoning traces for why some models were more vigilant than others. Most of the reasoning traces involved understanding the puzzle rather than reasoning about the advisor’s advice, although we note that based on increased token use in the malicious cases, models must be reasoning over advisors’ inputs to some degree.

---

### Author Response · Authors · 2025-12-03
**Summary for AC**

We thank all reviewers for their thoughtful feedback and constructive criticism. We are encouraged that reviewers found our dual focus on persuasion and vigilance to be "conceptually original" (5WYM), the Sokoban setup to be a “good, controllable way to test this” (NJnc), and that this is a “meaningful contribution toward understanding LLM robustness to external influence” (nSG7).

Based on the reviews, we identified four major themes: the choice of the Sokoban environment, the reliance on symbolic planners, the definition of our metrics, and the focus on LLM-LLM interactions instead of human-LLM interactions.

**1\. Generalization and Ecological Validity (Response to 5WYM, NJnc, nSG7)** A primary concern raised was the "toy" nature of Sokoban and whether findings here generalize to high-stakes domains like medicine or finance. While we agree that Sokoban does not capture the full complexity of human social cognition or high-stakes emotional persuasion, we emphasize that we chose this environment precisely because it allows us to isolate persuasion and vigilance from the confounding factors present in "realistic" settings.

Sokoban provides a "petri dish" for social cognition: it involves sequential decision-making (unlike single-turn persuasion tasks which are the norm in the literature), offers multiple failure modes (deadlocking vs. suboptimal planning), and allows for objective ground-truth evaluation. Our results should be viewed as a controlled window into model behavior, establishing a baseline for how persuasion, performance, and vigilance can be disentangled. We are excited for future work – motivated by our experiments here – to scale these paradigms to richer domains. We hypothesize the core finding (that capability does not equal vigilance) is robust, though this is a testable question for the broader research community to next explore.

**2\. Reliance on Symbolic Planners (Response to 5WYM, NJnc, nSG7, QCSx)** Several reviewers asked if our use of a symbolic planner for the "Advisor" creates an artificial separation or masks the model's true capabilities. The planner was originally intended to decouple the *ability to persuade* from the *ability to reason*. We wanted to test if a model could translate a plan into persuasion, regardless of whether it could generate the plan itself.

However, to address the concern regarding purely autonomous advisors, **we have run a new supplementary experiment (included in Appendix A.3) where Advisor models do not have access to a planner.**

* **Results:** We found broadly similar trends. Even without the planner, GPT-5 remains the most persuasive and vigilant model. Grok-4-Fast continues to be unpersuasive and unvigilant, despite strong baseline performance.
* **Implication:** While the absolute persuasion performance degrades slightly without the planner (as expected), the relative rank order and the dissociation between persuasion and vigilance remain consistent. This confirms that our findings are not artifacts of the symbolic planner.

**3\. Clarification of Metrics (Response to nSG7, QCSx)** We apologize for the confusion regarding the metric definitions and have rewritten Section 3.3.2 for clarity.

* **Normalization (Denominators):** In our persuasion and vigilance metrics (Eqns 2-5), the denominator represents only the trials where a signal is measurable. For example, if a player would have solved a puzzle unassisted, we exclude this trial from the "beneficial persuasion" denominator because we cannot measure if the advice helped. This prevents penalizing models for cases where persuasion was unnecessary.
* **Vigilance vs. Sycophancy (Response to NJnc):** We clarify that vigilance is distinct from non-sycophancy. A non-sycophantic model might simply ignore all input. Our vigilance metric requires a model to selectively *ignore* malicious advice while *following* beneficial advice.
* **Missing Data (Response to nSG7):** The missing data point for GPT-5 in Table 1 is because the model reached ceiling performance (100%) on the tasks unassisted; thus, there were no valid trials to measure "beneficial persuasion" improvement.

**4\. Human-LLM vs. LLM-LLM Interactions (Response to 5WYM, NJnc)** Regarding the inclusion of human participants or comparison to human-AI literature (e.g., Zhou et al., 2025): Our study is specifically scoped to LLM-vs-LLM interactions to understand LLM susceptibility to persuasion, which is becoming increasingly relevant as LLMs interact with content (or other LLM agents) online \[1\]. This also sets it apart from other studies looking at LLM persuasion, which mostly focus on LLMs persuasion of human participants.

\[1\] STÖCKL, A., AND NITU, J. Are AI agents interacting with online ads? arXiv preprint 2504.07112 (2025).

**We also added statistical significance tests to every claim within the paper as requested. With these changes and clarifications, we believe that we addressed the key reviewer concerns.**

---

### Meta-Review · Area_Chair_KBdK · 2026-01-07

**Summary:**

1. Toy environment: it remains unclear whether the findings generalize to real-world social or linguistic persuasion tasks. (5WYM, NJnC, nSG7,QCSx)
2. It remains uncertain how these results translate to human-AI scenarios. (5WYM)
3. Expectations for a deeper discussion about the mechanisms that drive persuasion effectiveness and vigilance. (5WYM)
4. The claim regarding the "dissociability" of puzzle solving, persuasion, and vigilance is not convincing given the agent settings. (NJnC, nSG7)
	4.1 The "advisor" models were given optimal solutions from a planner.
	4.2 The "vigilance" metric seems to be merely a measure of sycophancy; additionally, the significance of the vigilance score is questionable.
5. The definition of metrics is hard to follow. (nSG7, QCSx)

**Reviewer Concerns:**

1. (Still outstanding) The authors claim that the environment is intentionally tightly controlled to isolate research findings from confounding factors found in more "realistic" settings. However, this prevents the environment from being regarded as "scalable" or "rich." The authors also admit that scaling to richer domains is deferred to future work.
2. (Addressed) Human-AI interaction is out of the scope of this paper.
3. (Trivially Addressed) Some simple prompts may lead to "malicious awareness."

4.1 (Somehow Addressed) Added an additional experiment where access to a symbolic planner is not provided to the advisor model.

4.2 (Addressed) The authors explained that vigilance requires more than just a lack of sycophancy and modified their statements regarding the vigilance score.

5. (Still outstanding) Although the authors explained the details of the metric definitions, I agree with nSG7 and QCSx that the metrics are hard to follow.

**Reviewer Scores:**

I do not think any of the reviewers would change their scores, mainly because of the limited environment, which may significantly reduce the significance of the findings.

---

### Decision · Program_Chairs · 2026-01-26

Reject